# Feasibility and Validity of a Low-Cost Racing Simulator in Driving Assessment after Stroke

**DOI:** 10.3390/geriatrics5020035

**Published:** 2020-05-29

**Authors:** Jonathan Tiu, Annie C. Harmon, James D. Stowe, Amen Zwa, Marc Kinnear, Latch Dimitrov, Tina Nolte, David B. Carr

**Affiliations:** 1Department of Neurology, Washington University School of Medicine, St. Louis, MO 63110, USA; 2Department of Medicine, Washington University School of Medicine, St. Louis, MO 63110, USA; annieharmon@wustl.edu (A.C.H.); dcarr@wustl.edu (D.B.C.); 3Mid-America Regional Council, Kansas City, MO 64105, USA; jstowe@marc.org; 4sOnit, Inc., Potomac, MD 20854, USA; amenzwa@gmail.com; 5Thomas Dunn Learning Center, St. Louis, MO 63118, USA; marc@tdunn.org; 63 Peak Engineering, Toronto, ON M3C 3P3, Canada; latch.dimitrov@3peakintegration.com; 7Center for Clinical Studies, Washington University School of Medicine, St. Louis, MO 63110, USA; nolte.tina@wustl.edu

**Keywords:** return to driving, driving assessment, driving rehabilitation, racing simulation, driving simulation, stroke rehabilitation

## Abstract

There is a myriad of methodologies to assess driving performance after a stroke. These include psychometric tests, driving simulation, questionnaires, and/or road tests. Research-based driving simulators have emerged as a safe, convenient way to assess driving performance after a stroke. Such traditional research simulators are useful in recreating street traffic scenarios, but are often expensive, with limited physics models and graphics rendering. In contrast, racing simulators developed for motorsport professionals and enthusiasts offer high levels of realism, run on consumer-grade hardware, and can provide rich telemetric data. However, most offer limited simulation of traffic scenarios. This pilot study compares the feasibility of research simulation and racing simulation in a sample with minor stroke. We determine that the racing simulator is tolerated well in subjects with a minor stroke. There were correlations between research and racing simulator outcomes with psychometric tests associated with driving performance, such as the Trails Making Test Part A, Snellgrove Maze Task, and the Motricity Index. We found correlations between measures of driving speed on a complex research simulator scenario and racing simulator lap time and maximum tires off track. Finally, we present two models, using outcomes from either the research or racing simulator, predicting road test failure as linked to a previously published fitness-to-drive calculator that uses psychometric screening.

## 1. Introduction

For many individuals, driving is an integral activity of day-to-day life. Conversely, for individuals who undergo a stroke and transition into driving retirement, such a change in mobility is associated with poorer health outcomes, higher rates of depression, and lower rates of community integration [1,2,3]. Driving is a skill that requires the coordination of high-level cognitive and motor function to perform safely. When a patient has a stroke, it remains a challenge for the clinician to determine whether it is safe for the patient to return to driving [4,5,6,7]. There is a myriad of methodologies used to assess the safety of returning to driving after stroke. These include, but are not limited to, road tests and psychometric assessments [8,9].

The performance-based road test is often considered a reliable test to assess driving ability after stroke, and it is considered a standard criterion to which other tests with the same goal are often compared [10,11]. However, the road test can be costly, and patients’ access to these tests can be limited. Therefore, it has been a topic of interest to develop low-cost, easily accessible models predictive of success on a road test after stroke, which by extension aims to predict a safe return to driving [8]. Using a battery of clinical assessments and psychometric tests predictive of road test performance is one such approach, and our group has previously published a fitness-to-drive (FTD) calculator to predict the probability of road test failure using two brief psychometric screens that tap into processing speed, visuospatial skill, planning, and foresight [5].

High-end, research-based driving simulators have emerged as a viable supplement to assessing the safety of returning to driving after stroke [12]. These traditional research simulators are highly customizable platforms that can assess driving performance in a variety of contexts simulating real-life driving scenarios. While driving simulation is obviously different from real-life driving, evidence supports the use of research-based driving simulators to predict real-life driving performance, and they may be useful prospective predictors of driving performance in neurologically impaired adults [13,14,15,16,17,18]. Research-based driving simulators typically involve a life-size vehicle cab as well as a multiple monitor setup to simulate the wide field of view required in real life driving.

However, there are drawbacks to research simulators that may limit their accessibility. Research simulator units are costly, and higher-fidelity models often require budgets normally only available at the research institution level [19,20,21]. Driving rehabilitation in general, of which use of a research simulator may be included, requires multiple visits with health care professionals and is not typically reimbursed through usual insurance mechanisms.

Whereas research simulators aim to recreate the experience of community-based road driving, consumer-grade racing simulators emphasize fidelity to the experience of a motorsport driver on a competitive racing track [22,23]. Such racing simulators, which are targeted to automotive racing professionals and enthusiasts alike, offer enhanced physics models, immersive graphics rendering, and feature-rich telemetry all at very low cost. However, while both road driving and racing share several obvious skills, the two activities have some fundamental differences. Compared to road driving, automotive racing is conducted on wider tracks, with turns designed to physically challenge the driver in both solitary practice and in competitive scenarios where as many as 20 drivers may be competing simultaneously [24]. Race cars are also configured to perform at velocities and levels of acceleration/deceleration much greater than those seen in road cars [25]. Differences in cortical activation and eye movements in racers vs. non-racers have also been described, suggesting physiologically different adaptations to the two disciplines [26,27]. As such, it should be stressed that racing simulation is not meant to mimic road driving, but to offer a more strenuous driving experience. As a result, most racing simulators are limited in their ability to simulate street traffic and measure driver behavior in real world-based scenarios.

In spite of these differences, low-cost racing simulators possess another compelling feature: they provide feedback on driver performance using the same telemetric data and analysis techniques employed by real-world racing professionals [22]. Such feedback on performance from these low-cost simulators may have utility for future outpatient or home-based rehabilitation opportunities. For instance, the Assetto Corsa racing simulator can make use of a telemetry plug-in, the Assetto Corsa Telemetry Interface (ACTI), to harness a comprehensive array of sensors in gathering driver performance data [28]. The collected data may then be analyzed using statistical techniques commonly used in research, similar to the data extraction and reduction methods used for research simulators. Data can also be captured remotely over the internet.

One of the key advantages of the racing simulator is its low cost and flexibility for use at the individual consumer level. Racing simulators can run on desktops configured for computer gaming, on one or more monitors, and accept a number of consumer-grade force feedback steering wheels. Thus, a complete simulator system with only a few portable components can be assembled for under $1000 USD. A summary comparing custom-built research simulators and low-cost racing simulators is outlined in Table 1. The feasibility of using low-cost racing simulation has been explored in other clinical contexts, such as attention-deficit hyperactivity disorder [29]. Given these features, we aim to explore the feasibility of using a low-cost racing simulator as a tool in driving assessment after strokes [30].

Our aims for this observational-analytic pilot study were threefold. We first sought to determine the feasibility and tolerability (i.e., successfully completing simulator participation with minimal adverse events) of a low-cost racing simulator in a stroke patient cohort. We then sought to determine if parameters of research simulation and racing simulation correlated with clinical and psychometric characteristics in our cohort and, moreover, if parameters of the two simulators correlated between themselves. Finally, using our probability of road test failure calculator, we sought to determine if driving and racing simulator outcomes can each help reliably predict the risk of failure of on-road testing. We hypothesized that impaired driving simulator performance would be associated with performance on psychometric tests of processing speed, executive function, and visuospatial skills in our stroke sample.

The results of our study demonstrate that the use of consumer-grade racing simulators after minor stroke is well tolerated. Correlates between psychometric and physical (Snellgrove Maze Task [SMT], Trail Making Test Part A [TMT-A]) and Motricity Index [MI] of the right side) clinical assessments and several research and racing simulator outcomes are identified. We also present two models using parameters from either the research or racing simulator to predict a low- (≤25%) or high-risk (≥70%) probability of road test failure.

## 2. Materials and Methods

### 2.1. Sample

We employed the following inclusion criteria to our sample: active driver’s license; history of driving at least once per week prior to the stroke; ≥10 years driving experience; medical referral indicating clinical and/or imaging evidence of a stroke. Both active and non-active drivers were recruited, and prior racing experience was not used as a criterion for participation. The exclusion criteria were as follows: active severe depression (which may impair driving performance) [3]; history of severe motion sickness; visual acuity that does not meet state guidelines for driving (i.e., cannot be worse than 20/60 in both eyes); non-English speaking; major chronic unstable disease or condition (e.g., seizures); severe orthopedic/musculoskeletal impairments that would require extensive adaptive equipment to drive (e.g., mouth controls); severe visual or hearing impairments that interfere with the ability to perform study tests; sedating medications (e.g., new use of narcotics or anxiolytics within the past month, or chronic use that causes sedation which precludes safe driving). The majority of the participants were referred by occupational therapists or physicians. The rest were recruited from a research volunteer registry and were contacted directly by the research team. We recruited 41 adults with stroke for our pilot sample. We obtained the approval to conduct the study from the Washington University Human Research Protection Office (IRB ID #201506122).

### 2.2. Procedure

Participants underwent a research session consisting of demographic and background questionnaires, psychometric testing (paper-and-pencil as well as computer-based), clinical examination, and driving assessment (on both the research and racing simulators) [32]. Average session duration was 2.5 h (i.e., consent/questions: 20 min; psychometrics: 1 h; research simulator: 40 min; racing simulator: 20 min) including room transitions and equipment familiarization. Participants who requested adaptive equipment for the simulators (e.g., steering knob or left-foot accelerator) were permitted as long as the participant had prior training and experience with such devices.

We employed established psychometric tests typically adopted in clinical driving and research settings. We identified driving errors made on both types of simulators. These tests are thought to have value for identifying a baseline of performance from which the effectiveness of future training or rehabilitation interventions could be gauged [33].

### 2.3. Clinical and Psychometric Testing

#### 2.3.1. Clinical Background

Participants completed detailed health questionnaires to provide additional contextual details, including time since stroke (months), comorbidities, and medications. Diagnoses and medications associated with symptoms or side effects that potentially impair driving such as vision, alertness, cognition, or motor control, were totaled for each participant [34,35]. Thirty-nine chronic and acute conditions were identified as potentially impairing, including acute and general chronic conditions, as well as psychological, neurological, and sensory conditions. Acute medical conditions and symptoms included angina, dizziness/lightheadedness, drug or alcohol abuse, seizure disorders, syncope, and vertigo. General chronic conditions included arrhythmia, coronary artery disease, cancer, chronic heart failure (CHF), chronic obstructive pulmonary disease (COPD)/asthma, diabetes mellitus, hypothyroidism, morbid obesity, musculoskeletal (MSK)/chronic pain disorders, pacer, and sleep apnea. Psychological chronic conditions included attention deficit disorder (ADD), bipolar disorder, depression, generalized anxiety, schizophrenia, and chronic substance abuse. Neurological conditions included stroke, brain injury, cerebral palsy, dementia, multiple sclerosis, narcolepsy, Parkinson’s disease and spinal cord injury. Sensory limitations included cataracts, diplopia/visual field loss, glaucoma, hearing loss, macular degeneration, neuropathy, and retinopathy. The total number of potentially impairing conditions including stroke (1–39) were summed for each participant.

Participants additionally self-reported their medications, nineteen of which are considered potentially impairing to driving performance [34]. Individual medications types included: insulin, antihistamines, opioids, antidepressants (inclusive of psychiatric and pain-related indications, and including selective serotonin reuptake inhibitors, serotonin and norepinephrine reuptake inhibitors, tricyclic antidepressants, and monoamine oxidase inhibitors), muscle relaxants, benzodiazepines, barbiturates, hypnotics, and neurostimulants. Other agents included anti-dementia medications, anti-epileptics, anti-parkinsonian medications, anti-psychotics, proton pump inhibitors, and H2-receptor antagonists. Finally, immunosuppressive agents to treat cancer were included. The total number of potentially impairing medications (0–19) were summed for each participant.

#### 2.3.2. Clinical Exam

Participants completed several clinical measures to assess motor and neurologic impairment due to stroke. Participants were administered the Motricity Index [36] to assess for stroke-related motor impairment. Motor ability and gross motor coordination were assessed with the Rapid Pace Walk [37]. The assessment requires the participant to walk twenty feet; the amount of time to completion has been linked to driving outcomes [9]. A trained clinician administered the National Institute of Health Stroke Scale (NIHSS), a 15-item examination scale that assesses stroke-related neurologic impairment [38]. In assessing whether simulator performance was influenced by driving using the foot on the side affected by stroke (e.g., a participant with stroke affecting the right side using the right foot to operate the pedals), we used NIHSS scores on questions 5 and 6 to determine affected side. Participants were considered to have the right side affected if they scored points for right arm and/or leg (questions 5B and 6B) but scored 0 for left arm and/or leg (questions 5A and 6A); they were considered to have the left side affected if they scored points for the left side and scored 0 for the right side; finally, participants were considered to have both sides affected if they scored points for both the left and right sides.

#### 2.3.3. Vision Testing

The near visual acuity of participants was assessed using Sloan letters, and far visual acuity was assessed using the Early Treatment Diabetic Retinopathy Study (ETDRS, or LogMAR [Logarithm of the Minimum Angle of Resolution]) chart [39]. The contrast sensitivity of participants was assessed using the Pelli-Robson Contrast Sensitivity chart, which measures the ability to distinguish between descending increments of light vs. dark [40].

#### 2.3.4. Cognition

Patients were administered the Trail Making Test Part A (TMT-A), which assesses psychomotor processing speed and visuospatial ability [41]. The task requires participants to visually scan a page and connect numbers in order. Longer completion times indicate poorer processing speed and poorer visual attention abilities. Participants were administered the Snellgrove Maze Task (SMT) to assess executive functioning and visuospatial ability [42]. Longer completion times indicate poorer visuospatial ability and poorer executive functioning. Scores for these measures were entered into an algorithm to produce a probability of failing an on-road test [5].

### 2.4. Driving Simulation

#### 2.4.1. Research Simulator

For this study we used the STISIM M300WS research simulator from Systems Technology, Inc. (Hawthorne, CA, USA), which is equipped with three widescreen monitors conferring a 135° field-of-view, as well as a full-size driving compartment. The simulator can be customized to measure various parameters. A full array of outcome data can be captured if the system is preprogrammed to do so. We conducted a brief orientation session for each participant, in order to familiarize the participant to the system, as well as to combat simulator adaptation syndrome (SAS). We used four scenarios of increasing complexity and challenge to measure the driver’s performance and the ability to adhere to traffic safety regulations.

Before the start of each scenario, participants were instructed to maintain lane position and obey posted speed limits as much as possible, while responding to the demands of each drive. Scenario 1 (familiarization) involved a simple highway with gentle curves and light traffic to help participants familiarize themselves with the simulator. Scenario 2 (rural highway) involved a highway section with few buildings and diffuse traffic. Participants were told to pass slow vehicles in their lanes when they felt safe to do so, with up to 7 overtaking opportunities with the potential for collision. Participants’ ability to sustain attention was then tested with a long stretch (~60–90 s, depending on driving speed) without opportunities to pass. This scenario culminated in an event where participants were faced with oncoming vehicles in both lanes, forcing either collision or avoidance by leaving the roadway to pass on the right shoulder. Of all the research simulator scenarios, driver performance in Scenario 2 was thought to have among the highest clinical relevance to overall road driving safety: this scenario was designed to challenge participant attention, concentration and judgment by combining long stretches of highway speed driving with passing opportunities requiring calculated safety assessments, and the oncoming vehicle event requires brisk reaction time and prompt safe decision making (i.e., intentionally driving off the road) to avoid collision. Scenario 3 (construction) entailed a short but complex construction site course with a stationary obstacle collision opportunity. Scenario 4 (urban) comprised a dense, busy urban environment with traffic signals and pedestrians as well as vehicular collision opportunities. This scenario included a quick reaction event, where drivers who chose to drive through a programmed yellow traffic signal would suddenly be in the path of a bus running the light; the short notice for this vehicle collision event results in very short time-to-collision (TTC) values.

Research simulator measures included standard deviation of lateral position (SDLP; i.e., weaving) averaged through a scenario, speed through courses (minimum, maximum, average), braking (maximum degree of brake pedal depression), safety errors (e.g., collisions, off-road excursions, and traffic signal violations), serialization of vehicle control inputs during emergency maneuvering, time-to-collision (TTC), and eye movement parameters (not reported here) [27,43,44,45,46,47,48,49,50,51]. A pedal error was considered to be unintentional application of the wrong pedal (e.g., braking instead of accelerating) or unintentional application of both pedals simultaneously.

The six research simulator outcomes selected for the regression model indicated vehicular control, safe decision making, reasoning, planning, and reaction time. From Scenario 1 (familiarization), participants’ SDLP indicated vehicular control. From Scenario 2 we included maximum speed (as a general marker of adherence to speed limits), as well as a group of variables from the most challenging portion of the drive, where there are oncoming vehicles in both lanes: reaction time to oncoming vehicle in their lane (first control input after vehicles appear on screen, whether steering, brake, or accelerator input), collision (y/n) and road excursion (y/n); in this situation excursion was considered the most appropriate maneuver. Finally, TTC in Scenario 4 was included as a test of reaction speed in response to an unexpected stimulus (a car crossing a traffic light illegally).

#### 2.4.2. Racing Simulator

Our racing simulator utilized Assetto Corsa software (AC; developed by Kunos Simulazioni, Rome, Italy), an advanced racing simulator which is designed to run on consumer-grade PCs. AC has an engineered physics model believed to be highly accurate, realistic racetracks, and a high-fidelity audiovisual environment. AC comes prepackaged with many accurate car and track models. Car models that are licensed directly from the original manufacturers, and track models are acquired by laser scanning real-life race tracks. We augmented AC with an advanced data acquisition tool (Assetto Corsa Telemetry Interface, ACTI v1.1.1.0, Dimitrov, Toronto, ON, Canada) [28], and routed all telemetric data to the data analysis package, MoTeC i2 Standard (v1.1.2.0473, Melbourne, Victoria, Australia). Approximately 70 data channels can be exported to Excel or MATLAB for statistical analyses. We used a higher-end consumer-grade gaming PC setup, with a single, large widescreen monitor. For our consumer-grade racing steering wheel controller, we used an off-the-shelf Thrustmaster T500RS (Guillemot Corporation, Carentoir, France) force feedback steering wheel and pedals controller. The complete list and the cost of all components are included in Appendix A. The quantity of force feedback was turned down slightly from the default setting to accommodate for participants using adaptive steering equipment. Our machine was benchmarked at 157 frames per second (FPS) and provides a minimum of 60 FPS during simulation activities. No driving cab was used. We used the “Zwa method” of vehicle set-up and simulator configuration [22]. All participants used the FIAT Abarth 695SS vehicle, which was picked as having the slowest speed among the vehicles in the simulator.

To induce a more strenuous driving scenario, participants completed two courses of a racing simulator. Participants were instructed to drive as fast as possible, while maintaining control of the vehicle. As a practice course, we used Autodromo Nazionale Monza (aka, “Monza” [1966 layout]; SIAS S.p.A, Corso Venezia, Italy), because it features long straights joined together with a few gradual, undemanding corners. The test course (Silverstone Circuit [International Layout], Towcester, Northamptonshire, England) was used for final measurements. This track has the addition of a few sharp, challenging corners. Subjects drove at least one orientation lap to become familiarized with the simulator. Subjects then began sessions on each circuit from a standing (stationary) start, after which they completed two full laps with no interruption between laps. A drag race scenario was implemented at mid-point in the data collection (*n* = 21). Here, reaction time of each foot was measured for all subsequent participants using a dedicated reaction time test scenario (i.e., pedal response to a traffic light signal in a drag race setting).

Race simulator outcomes included lap time (including change in lap time between the two laps), SDLP (based on software-encoded racing line), number of tires that left the track surface (e.g., after losing control of the vehicle), throttle (maximum and mean), braking (maximum and mean), activation of the anti-lock braking system, and pedal reaction time for each foot when responding to an embedded traffic signal stimuli on a drag racing strip (i.e., time from illumination of green light to switching from brake to accelerator pedal application). “Preferred foot pedal reaction time” was assumed to be the right foot, except in cases of participants using adaptive left foot accelerator, in which case it was assumed to be the left foot. In contrast to capture of pedal error in the research simulator, whether the left or right foot was used for accelerator or brake pedals during racing scenarios was not collected explicitly and was left to participant preference.

The racing simulator outcomes selected for the regression model included lap 2 times, as well as the total number of tires that left the track in laps 1 and 2 for both the Silverstone and Monza courses. Participants were expected to use the initial lap more so to familiarize themselves with the layout and vehicle control, while the second lap would be more indicative of attention, sustained vehicle control, and wayfinding, i.e., more meaningful indicators of executive cognition and driving performance. Vehicular control was measured by the sum of the maximum numbers of tires off the track at one time for each lap of a course.

#### 2.4.3. Simulator Adaptation Syndrome

We made a concerted effort to reduce simulator adaptation syndrome (SAS, also known as simulator sickness) through a series of changes to the graphics settings of the research simulator (e.g., adjustment frame rate and simplification of background graphics assets to reduce visual detail) [52]. Our racing simulator used higher end components well above racing simulator requirements to ensure a frame rate of at least 60 FPS to reduce SAS. We defined SAS as follows: when a participant rated any symptom on the Simulator Sickness Questionnaire (SSQ) as “severe”; rated any two or more symptoms as “moderate”; or if the participant requested to discontinue testing due to symptoms [53].

### 2.5. Statistical Analyses

Means and standard deviations across study participants were computed for the descriptive analyses. Spearman correlation was used to identify bivariate correlations between clinical variables and research and racing simulator parameters. Univariate distributions were checked for normality and to identify outliers using the Tukey method (i.e., values above or below 1.5 times the interquartile range, IQR). To determine the effect of outliers, data were analyzed with the outliers, then again with outliers replaced with the nearest non-outlier values. A chi-squared test or independent samples *t*-test where appropriate were used to check for differences in research and racing simulator outcomes between those who used the affected side for their primary driving foot vs. those who used the unaffected side for their driving foot. We used binary logistic regression to develop two models comparing variables of research and racing simulator performance predictions of high or low probability of failing a standardized driving performance test, based on the fitness-to-drive calculator [5]. The outcome for both models is high (≥70%) and low (≤25%) probability of failing a standardized performance-based test. These cut-offs were determined based on the data distribution and clinically convenient reference points when counseling patients. Separate binary logistic regression models were created for racing simulator and research (STISIM) simulator variables to compare utility in predicting on-road performance. Analyses were performed using SPSS version 26.

## 3. Results

### 3.1. Sample Characteristics

The descriptive characteristics of our sample of 41 subjects with stroke are described in Table 2 (demographic characteristics) and Table 3 (clinical characteristics). Participants were, on average, older adults (63 years) with relatively minor and recent strokes. The mean time since stroke was 3.3 years (40 months), with relatively low NIHSS scores (3.27 on a scale of 0–42. Roughly 58% of participants identified as White, 39% Black, and 2% Asian. Nearly three-quarters of participants had never used video games prior to this study.

Table 4 summarizes results of cognitive and motor testing for our study sample. Average scores on tests that challenge high-level cognition (Trailmaking Test Part A [TMT-A] and Snellgrove Maze Task, [SMT]) showed little impairment in the sample overall. These two scores were used to generate individual probabilities of failing a driving performance road test. The histogram of these probabilities are displayed in Figure 1, indicating a largely low-risk sample with a bimodal distribution.

### 3.2. Simulator Results

#### 3.2.1. Participant Usage

During testing, 1 participant in this sample experienced simulator adaptation syndrome (SAS); the total sample SAS rate was 2.4%. A total of 6 participants used adaptive equipment on the research and racing simulators: 5 used a left foot accelerator, and 1 patient used a steering wheel spinner knob to assist with steering. Eight participants used the foot on their unimpaired side to operate the accelerator: 4 with right side impairment used their left foot, 4 with left side impairment utilized their right foot. Nineteen participants had right-side body impairment from the stroke but continued to operate the pedals with their right feet. Stroke side data was inconclusive for the remaining 14 participants. Overall, there were no differences in simulator performance measures between the groups using unimpaired side foot compared to impaired side foot acceleration (Appendix A). The single exception was Monza Lap 1 time, (t (20.237) = −2.265, *p* = 0.04), where the unimpaired subgroup included only 5 participants.

#### 3.2.2. Simulator Performance

Table 5 includes parameters of subject performance in the research simulator. Overall, participants adapted to the simulated driving environment well, with low average standard deviation of lateral position (SDLP) on the familiarization drive (scenario 1), indicating reasonable vehicular control. When faced with two oncoming cars at the end of scenario 2, the majority (70%) of participants experienced a vehicle collision. However, approximately half the sample (48.8%) made the most appropriate (if not always successful) choice to avoid collision, by driving off the right side of the road (road excursion). In line with scenario design intended to challenge reaction time, time-to-collision (TTC) in scenario 4 had a very short average (0.36 s) and narrow range.

Table 6 includes racing simulator performance outcomes which intend to show how well participants learned to navigate the courses (lap time) and control the racecar (tires off track), indicating their adaptation to the racing simulator system. Participants on average drove slower and took longer to navigate lap 1 than lap 2 on both racing courses. The average change was 23.3 s on Monza and 5.90 s on Silverstone. However, the differences were not statistically significant (Figure 2). Similarly, the sample’s average maximum number of tires off track were lower for the second lap for both courses (2.82 tires on Silverstone lap 1 vs. 2.29 lap 2; 3.03 tires on Monza lap 1 vs. 1.97 lap 2), however the difference did not reach statistical significance. Participants’ preferred foot pedal reaction time averaged under two seconds when measured using the racing simulator’s drag race function. While both right and left feet were tested for pedal reaction time, only the value for the primary driving foot was reported.

### 3.3. Binary Logistic Regression Models

#### 3.3.1. Hypothesis 1: Relationship of Clinical Psychometric and Motor Testing to Research and Racing Simulator Outcomes

Speeds on research simulator scenarios designed to test participants’ executive functioning to plan and adapt to on-road stressors (i.e., passing intermittently on a long stretch of roads, and navigating a complex construction zone) are significantly related to both cognitive and physical clinical measures (significant correlations are listed in Table 7, with complete correlations in Appendix A). For the research simulator, reaction time for the oncoming vehicle event in scenario 2 correlated with both cognitive and physical clinical measures, namely the Snellgrove Maze Task (SMT) and Motricity Index (MI). Racing simulator lap times correlated with both cognitive and physical measures, namely the Trail Making Test Part A (TMT-A) and Rapid Pace Walk (RPW).

#### 3.3.2. Hypothesis 2: Relationship of Racing Simulator to Research Simulator Outcomes

Significant correlations between the racing and research simulator outcomes are included in Table 8, with complete correlations included in Appendix A. Lap times on the racing simulator were significantly related to research simulator speeds in scenarios 2 (rural highway) and 3 (construction zone). Negative correlations were found between lower lap times on the racing simulator and faster speeds on the research simulator. The total maximum number of tires off track for both racing simulator courses correlated with maximum speed on research simulator scenario 2.

#### 3.3.3. Hypothesis 3: Relationship of Fitness-to-Drive Prediction to Simulator Outcomes

The results from binary logistic regression models indicate that variables of driving simulator telemetry were able to accurately classify participants as having a low (≤25%) or high (≥70%) probability of failing a road test, based on the fitness to drive probability calculator. Model 1 (*n* = 28; Table 9) correctly classified 92.9% of cases based on research simulator outcomes that measure vehicular control and executive function tests. The model more accurately predicted a low risk of failure (95.5%, 21/22) than a high risk of failure (83.3%, 5/6). The research simulator telemetry model explained 76.7% (Nagelkerke R^2^) of the variance in low or high risk of failing an on-road test. Notably, our model itself reached statistical significance (χ^2^ = 19.164, df = 6, *p* < 0.01), while individual predictors did not. Including or replacing outliers in Scenario 1 SDLP and Scenario 4 TTC did not meaningfully change any of the model estimates (Appendix A).

Similar results were found for the racing simulator model (Model 2), built around predictors that measure vehicular control and executive cognition. Model 2 (*n* = 26; Table 10) correctly classified 92.3% of cases, with more accuracy in the low-risk group (100%, 21/21) than the high-risk (60%, 3/5), and explained just over half (Nagelkerke R^2^ = 53.4%) of the variance in low or high risk of failing an on-road test. As with Model 1, while Model 2 itself reached statistical significance (χ^2^ = 10.723, df = 4, *p* < 0.05), the individual predictors did not. Including or replacing outlying values in Monza Lap 2 did not meaningfully change any of the model estimates (Appendix A).

## 4. Discussion

Traditional research-based driving simulators aim to mimic realistic and immersive driving environments (e.g., driving cab with bucket seats, multiple screens with wide viewing angles, and real-world traffic scenarios), but their size and cost can be prohibitive in a clinical setting. Racing simulators, on the other hand, are available at the consumer level and are thus cheaper and more accessible. However, little empirical evidence is available to discern the validity, adaptability, and tolerability of these racing simulators to assess or rehabilitate medical driving impairments, including stroke. In this exploratory study, we assessed how well adults with minor stroke tolerated racing simulation, and we also compared driving performance outcomes on the two types of driving simulators. We also determined that simulator performance outcomes could accurately group participants into low or high risk of failing an on-road skills test, as determined by our previously published fitness-to-drive calculator.

The research and racing simulator were both well tolerated overall, with few symptoms of SAS syndrome in this sample (only 1 of 41 patients experienced SAS symptoms). During our investigation, participants were noted to provide spontaneous feedback on the racing simulator. This feedback was not collected in a systematized manner, although it was passively observed that participants found the controls realistic, the graphics attractive and convincing, and vehicle sounds and physical responses reminiscent of real driving. A few participants disagreed with steering feel (e.g., “unrealistic,” “challenging,” and “felt heavy”). This complaint about heavy steering feel may be due to the fact that the participants were unfamiliar with the steering feel of high-performance race cars, which the racing simulator seeks to accurately recreate. Future work can attempt to collect feedback in a systematized way to better understand a participant’s experiences with the racing simulator, which might also serve to refine future racing simulator assessment protocols. Furthermore, our racing simulator did not have a driving cab—such a feature may enhance simulation immersion but is not typically available to the average consumer. While literature on construct validity of research simulators is already established, additional studies will be needed to establish the construct validity of low-cost racing simulators [43,54,55].

Importantly, performance measures from the racing simulator were significantly related to clinical tests of physical and cognitive function, emphasizing the potential ability of consumer-grade racing simulators to flag stroke patients for driving impairment. In our sample, Trail Making Test A (TMT-A) correlated with lap 2 times on both the Monza and Silverstone racing simulator courses. As a primary component of the fitness-to-drive calculator [5], which calculates the probability of failing a road-based performance test, TMT-A is one of the best clinical predictors of driving [56]. This link to an established clinical correlate of driving performance suggests that lap times—which are essentially measures of general racing performance—may have some significance when using the racing simulator as a driving assessment tool. Physical measurements such as limb and grip strength also correlated with racing performance in terms of vehicular control, although not consistently between courses. Larger samples are needed to see if these relationships hold, as well as other racing performance outcomes associated with fitness to drive.

Interestingly, maximum speed in scenario 2 of the research simulator correlated with fast lap times in the racing simulator, as well as measures of vehicular control (max # of tires off track) in both tracks of the racing simulator. It is important to highlight that while maximizing speed is important in the racing simulator, speed limits and safe adherence to traffic regulations were emphasized in the research simulator. With that said, few participants exceeded the speed limit in research simulator scenario 2. It makes sense that being able to navigate the racing simulator courses at high speed with a minimum of technical errors correlates with safe and effective driving on the complex research simulator scenarios. Of note, other than in the drag race scenario in the racing simulator, we did not collect data on use of left or right foot for each pedal in the racing simulator. It is generally noted that most race drivers—even at amateur levels—use “left-foot braking” [22]. It would be valuable to characterize this behavior during racing in a stroke population as it pertains to lesion side and presence of any hemiparesis.

We were able to create two models—using either the research simulator or the racing simulator alone—for predicting patients at low- and high-risk of road-test failure as determined by our fitness-to-drive calculator. Both of our simulator models reached statistical significance, but individual components of each model did not. Compared to our research simulator model, our racing simulator model was less strongly predictive of road test failure; nonetheless, the racing model was able to detect 3/5 high risk drivers. The strength of both of our models may be an artifact of our small sample size. It may be an area of interest for future larger studies to compare performance on different racetracks and conducting sessions with an increased number of laps.

The findings of our small pilot study bring initial support for the idea that racing simulation may have the potential for clinical use in assessing driving fitness after minor stroke, but more data is needed before this practice can be confidently adopted into the clinical setting. On a practical level, for the patient with minor stroke who does not have ready access to driving simulation or driving rehabilitation resources, clinicians might be able to suggest the installation of a low-cost racing simulator at the patient’s home as a tool for assessing the safety of returning to driving after stroke. However, it ought to be stressed once again that racing is not meant to be a 1:1 analogue to road driving, but rather a modality aimed at providing a strenuous driving environment for which skills may correlate with road driving. Larger future studies are necessary to validate our results, to explore the transfer of skills between racing and research simulation, and most importantly, to determine direct correlations of racing simulation to the actual road. One of the strengths of our study was collecting simulator data previously shown to have significance. There are numerous simulated driving behaviors that are correlated with real-world driving performance such as motor vehicle crashes or performance-based road tests and which have been the subject of previous reviews. Vehicle speed, speed variability (standard deviation of speed), SDLP, brake reaction time, and TTC are frequent outcomes measured in driving simulation [43,44,45,46,47,48]. In our study, we were able to obtain the majority of these measures, or close proxies, during racing simulation.

Another of the strengths of our study was the relative simplicity of our protocol using widely available psychometric and motor tests as well as a popular racing simulator. This helps the reproducibility of our methods for future, related projects. Of note, another advantage of a racing simulation using popular simulator software is the opportunity for user-made modification, which are often freely available for download on enthusiast community websites. Following the completion of this study, there has been development of modifications for Assetto Corsa which add the feature of two-way highway traffic; such modifications may increase the relevance of these racing simulators to driving assessment and rehabilitation [31].

Our pilot study had a number of limitations. Firstly, we had a mixed sample of drivers and non-drivers. Recent driving experience may influence simulation performance and larger studies may be able to assess subgroup analyses based on whether participants are actively driving. In a related way, while it was not wholly surprising that none of our subjects had racing experience, future work ought to screen for this point. Secondly, our model utilized a fitness-to-drive calculator to indirectly predict road test performance. Future work that directly compares simulator outcomes to road test performance and/or prospective crash data will enhance validity measures. Additionally, we did not control for prior gaming experience. The majority of our sample had never played video games, while 5 (12%) of our subjects played daily. Prior gaming experience and transfer across racing simulators may influence simulator performance, with unclear transfer to real-world driving performance [57]. Furthermore, our pilot study featured a small sample size. Recruiting a larger sample would improve the accuracy of the predictions and also reduce Type II error.

Finally, we did not emphasize capture of stroke location; however, we were able to use chronic NIHSS as an indirect measure to approximate stroke lesion side. Obviously, stroke locations and sizes are heterogeneous and would impact performance on simulator testing. However, in our study most of our participants had very low NIHSS scores, and were either driving or had an interest in returning to driving. We additionally found a low occurrence of neglect in our sample based on NIHSS, although the sensitivity of the NIHSS for capturing this may not be great [58]. Future work could explore the impact of racing simulator use in more severe strokes.

In addition, we found nearly no performance difference between those driving with the affected foot compared to those driving with the unaffected foot. The sample’s low NIHSS score average may suggest that stroke severity was unlikely to be a major factor in this sample, thereby limiting left-/right-sided stroke differences in our sample. While this is an intriguing finding, the limitations of the data and sample size do not allow us to draw further implications. Unfortunately, only 8 participants drove with their unimpaired foot (*n* = 8; 4 right-body-affected participants who used a left foot accelerator, and 4 left-body-affected participants who continued to drive with their right foot), diminishing the statistical estimations. Nonetheless, lesion side is vital information that should be collected in future studies on racing simulation and stroke.

In conclusion, low-cost, high-fidelity racing simulation has matured in recent years, and our data support the rigorous study of this technology as a potential assessment tool to help individuals with minor stroke return to safe and active driving. If installed at participants’ homes, patients could minimize visits or return trips to a health care center, which may serve as an especially valuable shift for those already compromised in safe transportation.

## Figures and Tables

**Figure 1 geriatrics-05-00035-f001:**
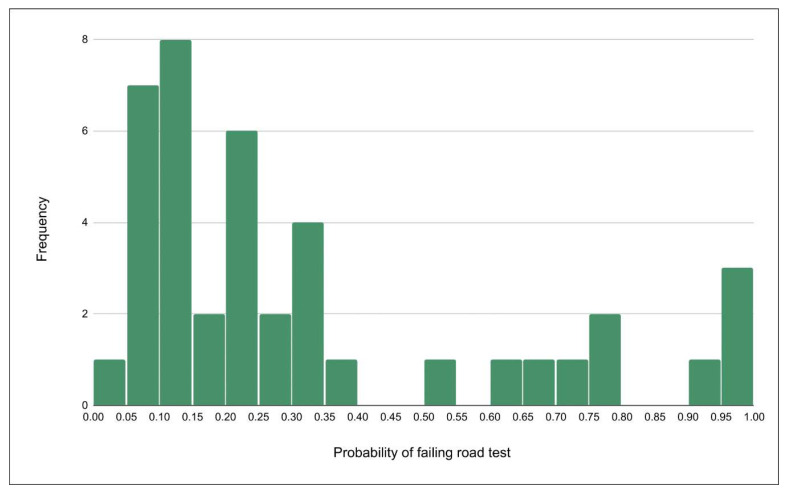
Histogram showing participants’ probability of failing a road-based driving test based on the fitness-to-drive (FTD) calculator for stroke patients. The FTD calculator utilizes psychometric testing scores (Trail Making Test Part A and Snellgrove Maze Task) and a stroke-specific algorithm to estimate the probability of failing the road test. While most participants had a higher probability to pass than fail a road test (scores below 50%), the bimodal distribution show potential to identify both low- (≤25%) and high-risk (≥70%) stroke patients on either end of the scale.

**Figure 2 geriatrics-05-00035-f002:**
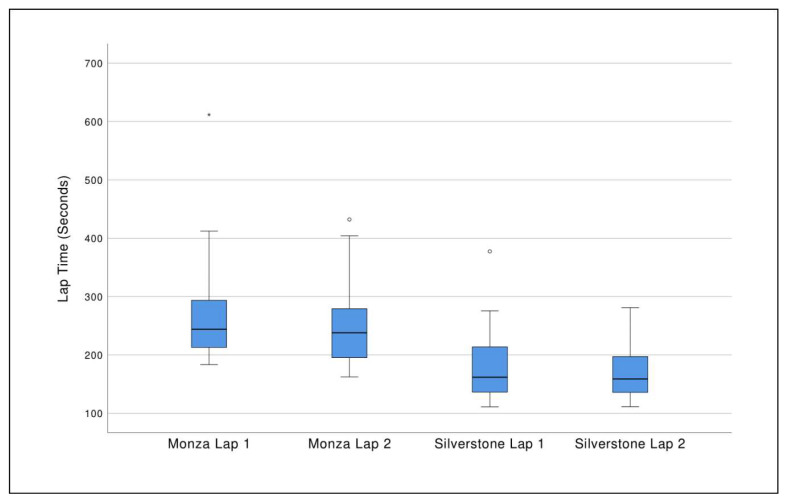
Boxplots showing participants racing simulator lap times (in seconds) for the Monza and Silverstone racetracks in Assetto Corsa. While participants decreased their second lap time on both courses on average, these changes did not reach statistical significance. A circle (○) denotes an outlier (1.5 times interquartile range) and an asterisk (*) denotes an extreme outlier (3 times interquartile range).

**Table 1 geriatrics-05-00035-t001:** Comparison of research simulator and racing simulator features.

Feature	Research Simulators	Racing Simulators
Cost	Moderate to high: can cost $15,000–$100,000 USD for cab, computer, and software packages	Low: software often available for $60 USD or less; force feedback steering wheels available for $100–600 USD; can run on personal computers approximately $500–2000 USD; cab/chair setups optional and variable in price
Physics modeling	Poor to fair	Superior
Graphics fidelity	Poor to moderate	Variable depending on computer hardware, but generally superior
Audio fidelity	Poor	Superior
Crash-related measures	Collisions, time-to-collision, traffic regulation violations	Number of tires off track, loss of control events
Modifiability	Full	Partial
Programmable traffic scenarios	Fully programmable	At time of study, limited to interaction with artificial intelligence (AI) competitors, or online gaming community; subsequent user-made traffic modifications have been under development [31]
Data extraction	Dependent on software; some allow data recapture using session replay, with others requiring pre-programmed data extraction	Most generally offer continuous data capture; recapturing data possible using session replay
Ease-of-use for clinician	Requires some training	Emphasis placed on intuitive, point-and-click interfaces
Primary troubleshooting and support	Manufacturer and colleagues	Online communities and forums
Evidence-based research in medically impaired drivers	Plethora of studies in the literature	Paucity of studies in the literature

**Table 2 geriatrics-05-00035-t002:** Demographic characteristics of the study sample.

Variable	Mean (SD)
Age	63.3 (11.66)
Female gender (Yes)	*n* = 20 (48.8%)
Currently Driving (Yes)	*n* = 17 (41.5%)
*Race*	
White	*n* = 24 (58.5%)
Black	*n* = 16 (39.0%)
Asian	*n* = 1 (2.4%)
*Level of education*	
High School, GED, or less	*n* = 5 (12.2%)
Some College/Vocational/Tech	*n* = 13 (31.7%)
2-Year Degree (Associates)	*n* = 3 (7.3%)
4-Year Degree (BA, BS)	*n* = 8 (19.5%)
Advanced or Professional Degree	*n* = 12 (27.3%)
*Video game experience*	
Never	*n* = 30 (73.2%)
Daily	*n* = 5 (12.2%)
Missing	*n* = 6 (14.6%)

GED = General Education Development degree, BA = Bachelor of Arts, BS = Bachelor of Science.

**Table 3 geriatrics-05-00035-t003:** Clinical characteristics of study sample.

Variable (Score Range)	Mean (SD)	Range
PHQ-9 (0 to 27)	2.34 (3.77)	1–15
NIHSS (0 to 42)	3.27 (2.31)	0–9
Any visual field impairment, NIHSS question 3	*n* = 7 (17.1%)	---
Any extinction/inattention, NIHSS question 11	*n* = 3 (7.3%)	---
Stroke affecting left side	*n* = 20 (48.8%)	---
Stroke affecting right side	*n* = 4 (9.8%)	---
Stroke affecting bilateral sides	*n* = 3 (7.3%)	---
Stroke affecting unknown side	*n* = 14 (34.1%)	---
Time since stroke (months)	36.89 (47.54)	1–196
ESS (0 to 24)	4.00 (3.37)	0–13
HHIE-S (0 to 40)	1.17 (2.61)	0–10
Sloan letters (8 M to 0.2 M)	0.98 (0.25)	0.50–1.60
EDTRS chart (1 to −0.3)	0.96 (0.29)	0.25–1.25
Contrast Sensitivity (2.0 to 1.0)	1.69 (0.20)	1.05–2.10
PDI Medical Conditions (including stroke; 1 to 39)	3.22 (2.12)	1–10
PDI Medications (0 to 19)	1.37 (1.59)	0–7

PHQ-9 = Patient Health Questionnaire-9, NIHSS = National Institutes of Health Stroke Scale, ESS = Epworth Sleepiness Scale, HHIES = Hearing Handicap Inventory for the Elderly, EDTRS = Early Treatment Diabetic Retinopathy Study, PDI = Potential Driver Impairing.

**Table 4 geriatrics-05-00035-t004:** Results of cognitive and motor testing of study sample.

Variable (Score Range)	Mean (SD)	Range
TMT-A (seconds)	48.10 (24.16)	24.96–118.81
SMT (seconds)	39.39 (12.00)	22.13–73.08
Rapid Pace Walk (seconds)	9.32 (5.06)	4.18–32.15
*Motricity Index (points)*		
Right	88.35 (12.92)	43.5–100
Left	92.81 (8.01)	67.5–100
*Jamar Grip Strength (Kg)*		
Right	21.50 (13.07)	0.0–52.0
Left	25.30 (11.67)	0.0–50.0

TMT-A = Trail Making Test Part A, SMT = Snellgrove Maze Task.

**Table 5 geriatrics-05-00035-t005:** Study sample performance in the traditional research-based driving simulator.

Variable (Score Range)	Mean (SD)	Range
*Scenario 1: Highway with gentle curve and light traffic*		
SDLP (meters)	1.75 (1.11)	0.36–6.00
*Scenario 2: Rural highway with overtaking opportunities*		
Mean speed (MPH)	47.19 (12.88)	20.30–71.29
Max speed (MPH)	53.82 (10.43)	29.47–73.54
Collision (Yes)	*n* = 29 (70.7%)	---
Pedal error (Yes)	*n* = 15 (36.6%)	---
Road excursion (Yes)	*n* = 20 (48.8%)	---
*Scenario 3: Construction site*		
Mean speed (MPH)	34.98 (9.76)	15.24–51.37
*Scenario 4: Busy urban environment*		
Time-to-collision (seconds)	0.36 (0.27)	0.04–1.46

SDLP = standard deviation of lateral position, MPH = miles per hour.

**Table 6 geriatrics-05-00035-t006:** Study sample performance in the low-cost racing simulator.

Variable (Score Range)	Mean (SD)	Range
*Course 1 (Silverstone Circuit)*		
Lap 1 time (seconds)	177.66 (56.38)	111.01–377.40
Lap 2 time (seconds)	171.18 (46.32)	111.21–280.80
Lap 2 minus lap 1 time (seconds)	−5.90 (36.61)	−173.38–50.65
Lap 1, max # tires off track	2.82 (1.82)	0–4
Lap 2, max # tires off track	2.29 (1.99)	0–4
Total max # tires off track	5.08 (3.12)	0–8
*Course 2 (Monza Circuit)*		
Lap 1 time (seconds)	267.48 (80.24)	183.50–611.66
Lap 2 time (seconds)	244.85 (65.21)	162.25–432.12
Lap 2 minus lap 1 time (seconds)	−23.33 (63.15)	-218.05–176.52
Lap 1, max # tires off track	3.03 (1.59)	0–4
Lap 2, max # tires off track	1.97 (1.95)	0–4
Total max # tires off track	4.94 (2.88)	0–8
*Drag Racing Scenario*		
Preferred foot pedal reaction time (seconds)	1.97 (0.55)	1.15–3.09

**Table 7 geriatrics-05-00035-t007:** Significant spearman correlations of clinical psychometric and motor testing to research and racing simulator outcomes.

Clinical Test	r_S_	*p*
*Trail Making Test Part A*		
Silverstone lap 2 time	0.346 *	*p* = 0.04
Monza lap 2 time	0.452 **	*p* < 0.01
*Snellgrove Maze Task*		
Rural scenario passing reaction time	0.337 *	*p* = 0.03
*Rapid Pace* Walk		
Monza lap 2 time	0.394 *	*p* = 0.02
*Motricity Index, right side*		
Rural scenario mean speed	−0.476 **	*p* < 0.01
Rural scenario max speed	−0.352 *	*p* = 0.03
Rural scenario passing reaction time	−0.588 **	*p* < 0.01
Construction scenario mean speed	−0.460 **	*p* < 0.01
Silverstone total max # tires off track	−0.357 *	*p* = 0.03
Silverstone lap 2 max # tires off track	−0.325 *	*p* < 0.05
*Jamar grip strength, left side*		
Monza total max # tires off track	0.374 *	*p* = 0.03
Monza lap 2 max # tires off track	0.343 *	*p* = 0.04
Drag race preferred foot pedal reaction time	−0.460 *	*p* = 0.04

* *p* < 0.05; ** *p* < 0.01.

**Table 8 geriatrics-05-00035-t008:** Significant Spearman correlations of racing simulator to research simulator outcomes.

Racing Simulator Variable	r_S_	*p*
*Silverstone lap 1 time*		
Rural scenario max speed	−0.474 **	*p* < 0.01
Construction scenario mean speed	−0.336 *	*p* = 0.04
*Silverstone lap 2 time*		
Rural scenario max speed	−0.529 **	*p* < 0.01
Construction scenario mean speed	−0.417 *	*p* = 0.01
*Silverstone total max # tires off track*		
Rural scenario mean speed	0.396 *	*p* = 0.01
*Silverstone lap 2 max # tires off track*		
Rural scenario mean speed	0.521 **	*p* < 0.01
Rural scenario max speed	0.454 **	*p* < 0.01
Rural scenario passing reaction time	0.424 **	*p* < 0.01
Construction scenario mean speed	0.340 *	*p* = 0.04
*Monza lap 1 time*		
Rural scenario mean speed	−0.379 *	*p* = 0.02
Rural scenario max speed	−0.604 **	*p* < 0.01
Construction scenario mean speed	−0.611 **	*p* < 0.01
Urban scenario TTC	0.728 **	*p* < 0.01
*Monza lap 2 time*		
Rural scenario max speed	−0.490 **	*p* < 0.01
Construction scenario mean speed	−0.492 **	*p* < 0.01
Urban scenario TTC	0.495 **	*p* < 0.01
*Monza total max # tires off track*		
Rural scenario max speed	0.335 *	*p* < 0.05
*Drag race preferred foot pedal reaction time*		
Highway scenario SDLP	0.440 *	*p* < 0.05

TTC = time-to-collision, SDLP = standard deviation of lap position. * *p* < 0.05; ** *p* < 0.01.

**Table 9 geriatrics-05-00035-t009:** Research simulator variables in the binary logistic regression Model 1.

Low- Versus High-Risk Fitness-to-Drive Screen									
								95% CI
		B	SE	Wald	df	Sig.	Exp(B)	Lower	Upper
Step 1	Scenario 1 SDLP (ft)	−8.932	7.868	1.289	1	0.256	0.000	0.000	657.311
	Scenario 2 maximum speed (MPH)	−0.122	0.098	1.556	1	0.212	0.885	0.731	1.072
	Scenario 2 reaction time (sec)	1.529	0.908	2.837	1	0.092	4.612	0.799	27.312
	Scenario 2 road excursion (y/n)	2.583	8.160	0.100	1	0.752	13.235	---	---
	Scenario 2 Collision (y/n)	0.336	7.890	0.002		0.966	1.399	---	---
	Scenario 4 TTC	−15.118	10.044	2.266	1	0.132	0.000	0.000	96.305
	Constant	13.741	13.236	1.078	1	0.299	928273.551	---

CI = confidence interval, SDLP = standard deviation of lap position, MPH = miles per hour, TTC = time-to-collision.

**Table 10 geriatrics-05-00035-t010:** Racing simulator variables in the binary logistic regression Model 2.

Low-Versus High-Risk Fitness-to-Drive Screen									
								95% CI	
		B	SE	Wald	df	Sig.	Exp(B)	Lower	Upper
Step 1	Monza Lap 2 Time (sec)	0.027	0.02	1.88	1	0.17	1.028	0.988	1.069
	Silverstone Lap 2 Time (sec)	0.009	0.02	0.14	1	0.70	1.009	0.964	1.056
	Monza Total Max Tires Off Track	−0.067	0.282	0.06	1	0.81	0.935	0.537	1.626
	Silverstone Total Max Tires Off Track	−0.245	0.245	1.00	1	0.32	0.783	0.484	1.264
	Constant	−9.2	4.30	4.81	1	0.03	0.000		

CI = confidence interval.

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
