# Peer review of "Feasibility and Validity of a Low-Cost Racing Simulator in Driving Assessment after Stroke"

_geriatrics, 2020, doi:10.3390/geriatrics5020035_

Round 1

Reviewer 1 Report

Title: Feasibility and Validity of a Low-cost Racing Simulator in Driving Rehabilitation after stroke.

The study compared the driving performance in racing and research grade driving simulator in 41 individuals with stroke. The results demonstrate correlations between selected driving outcomes in both simulators. In addition, moderate correlations were found between driving outcome in both simulators with psychometric tests. A binary logistic model was used to predict fitness to drive (as calculated with psychometric screening variable previously developed by the authors) using racing and research driving simulator outcomes.

The pilot study is well designed. However, a few key methodological concerns are noted. The primary one concerns the prediction model applied to predict the fitness to drive.

  1. Since this study is comparing the fidelity of racing simulator outcomes with the research simulator outcomes, it is more appropriate to run separate logistic regressions for racing and research simulator outcomes. Further, the Statistical Analysis section provides insufficient information on type of binary logistic method used and the rationale for selecting the specific method.
  2. Both right and left side impaired individuals were recruited. However, authors have combined the results for right or left foot driver without considering the side of the impairment.

Specific Concerns

Title: The study examines driving performance not driving rehabilitation after stroke. Therefore, the title would be more accurate if the word ‘assessment’ was used in place of rehabilitation.

Abstract: Please clarify which specific research simulator outcomes were related to lap time in racing simulator.

Introduction: Table 1: It is not entirely accurate that the research simulators must preprogram data extraction because research simulator such as NADSmini allow data extraction using session replay after the data collection.

Methods:

  1. How was the feedback on consumer-grade racing simulator collected? Provide the precise method for data collection, specific questions asked and analysis on feedback. Without this information the author’s conclusion that “face validity of the racing simulator appeared high” is unsupported and should be eliminated.
  2. The authors recruited both right and left side affected drivers. While this improves the ecological validity of the findings, it is highly probable that individuals driving with the affected left had more impaired performance than those driving with the less affected side. Please explain, how did the performance on driving outcomes vary in individuals who drove with the affected side versus less affected side ?
  3. For the research simulator, please clarify what is implied by the driving outcome ‘maximum braking achieved”?
  4. What was the rationale for considering the scenario 2 to have highest clinical relevance to overall driving safety ?
  5. For the racing simulator, were both feet used for driving? It is unclear how and why the reaction time of both feet were measured.
  6. The statistical analysis section lacks information on binary logistic method which (i.e., forward/backward Likelihood ratio/conditional) was used and the rationale for selecting the specific method.
  7. Most importantly, the rationale for the adding variable from research and racing simulator in the same model are unclear. The study is comparing the fidelity of racing simulator outcomes with the research simulator outcomes. Therefore, it is more appropriate to run separate logistic regressions for racing and research simulator outcomes.
  8. Table 2: What are the numbers in parenthesis where the authors indicate the ‘n’ on variables?
  9. Table 3: Please provide the number of individuals that drive with the affected and unaffected leg.
  10. Table 6: the right and left foot pedal reaction times are provided. Were these influenced by whether the affected or less-affected side was used for driving ?
  11. Hypothesis 3: What was the rationale for choosing specific driving outcomes (lap 2 times, maximum brake percentage) among all other outcomes to enter in the binary logistic regression?
  12. Table 9: Please provide the 95% CI for Exp (B). 

Discussion

  1. The author’s conclusion that “face validity of the racing simulator appeared high “appear to be unsupported by results shown.
  2. The results on the correlation between cognitive and motor tests with brake pedal reaction time were not found in the results section. Please provide these or eliminate the discussion on these results.
  3. The authors state that the lap 2 times on racing simulator may be potentially significant in predicting fitness to drive. Please explain the directional impact of lap 2 time on the prediction of “fitness to drive”  and its implication on driving assessment in clinics ?

Author Response

Dear editors of Geriatrics Aging and Driving Special Issue,

Thank you for your insightful comments and your recommendations for major revisions to our manuscript, “Feasibility and Validity of a Low-Cost Racing Simulator in Driving Rehabilitation after Stroke”. Please find attached our response to the reviewers detailing the revisions, along with a ‘Track Changes’ version of our manuscript, now re-titled, “Feasibility and Validity of a Low-Cost Racing Simulator in Driving Assessment after Stroke”. 

We hope you agree that the manuscript is improved, and hope you consider once again our revised work for publication. We also thank you for your generous extension in light of our team’s clinical duties related to the COVID-19 pandemic.

We are very grateful for, and we include individualized responses to, our first reviewer’s comments in the attachment. Our comments are highlighted in red color for your convenience.

Sincerely,
Jonathan Tiu, MD

Reviewer 2 Report

see attached document

Author Response

Dear editors of Geriatrics Aging and Driving Special Issue,

Thank you for your insightful comments and your recommendations for major revisions to our manuscript, “Feasibility and Validity of a Low-Cost Racing Simulator in Driving Rehabilitation after Stroke”. Please find attached our response to the reviewers detailing the revisions, along with a ‘Track Changes’ version of our manuscript, now re-titled, “Feasibility and Validity of a Low-Cost Racing Simulator in Driving Assessment after Stroke”. 

We hope you agree that the manuscript is improved, and hope you consider once again our revised work for publication. We also thank you for your generous extension in light of our team’s clinical duties related to the COVID-19 pandemic.

We are very grateful for, and we include individualized responses to, our second reviewer’s comments in the attachment. Our comments are highlighted in red color for your convenience.

Sincerely,
Jonathan Tiu, MD

Round 2

Reviewer 1 Report

Few minor concerns remain.

  1. The lack of information on the affected side after stroke: The authors report that “NIHSS are very low and therefore stroke severity was unlikely to be a major factor, and thereby limiting left-/right sided stroke differences in the sample.” It is not surprising that NIHSS score were low given that driving is a complex task and perhaps can be most effectively performed by minor or moderately severe stroke survivors. However, despite a small NIHSS, the participants may have unilateral motor impairments, even though minor, that can affect driving. Moreover, individuals driving with the affected leg may have differential deficits compared to those driving with unaffected leg.

In the absence of the information about the lesioned hemisphere, the mean motor leg and arm scores for left and right side from the NIHSS score, can be used to determine the more affected and the less affected sides. This information can stratify stroke subjects into those driving with affected leg versus those driving with unaffected leg for conducting a secondary analysis.

  1. Further, the results presented in the manuscript were based on minor stroke subjects NIHSS (1-4). Perhaps, driving in racing simulator under strenuous conditions may not be possible for more severe strokes. Therefore, it is important to clarify and acknowledge in the abstract and discussion sections that the results are applicable only in minor stroke and cannot be generally applied to all stroke survivors.

  1. Figures 1 and 2 look identical and are difficult to interpret. Please revisit and provide adequate explanations for both figures alone with detailed figure captions.

Author Response

Dear editors of Geriatrics Aging and Driving Special Issue,

Thank you for your insightful comments and your recommendations for minor revisions to our manuscript, "Feasibility and Validity of a Low-Cost Racing Simulator in Driving Assessment after Stroke". Please find attached our response to the reviewers detailing the revisions, along with a "Track Changes" version of our manuscript which has now undergone minor second revisions. We hope you agree that the manuscript is improved, and hope you consider once again our revised work for publication. We are very grateful for, and we include individualized responses to, our first reviewer’s comments below. Our comments are highlighted in red color for your convenience.

Sincerely,
Jonathan Tiu, MD

-------

Reviewer #1 comments

Few minor concerns remain.

  1. The lack of information on the affected side after stroke: The authors report that “NIHSS are very low and therefore stroke severity was unlikely to be a major factor, and thereby limiting left-/right sided stroke differences in the sample.” It is not surprising that NIHSS score were low given that driving is a complex task and perhaps can be most effectively performed by minor or moderately severe stroke survivors. However, despite a small NIHSS, the participants may have unilateral motor impairments, even though minor, that can affect driving. Moreover, individuals driving with the affected leg may have differential deficits compared to those driving with unaffected leg.

In the absence of the information about the lesioned hemisphere, the mean motor leg and arm scores for left and right side from the NIHSS score, can be used to determine the more affected and the less affected sides. This information can stratify stroke subjects into those driving with affected leg versus those driving with unaffected leg for conducting a secondary analysis.

We thank you for this important point. We have performed statistical analyses comparing participants who drove using the side affected by stroke and those who drove using the side unaffected by stroke. This variable was determined by analysing the NIHSS scores for left and right sided drift, and is outlined on lines 185-192. The statistical tests used are outlined in our methods on lines 318-321. The results of this secondary analysis is reported on lines 363-370 and a new Table S2 in the Appendix. We found minimal differences between these groups and comment on this in our discussion on lines 555-563.

  1. Further, the results presented in the manuscript were based on minor stroke subjects NIHSS (1-4). Perhaps, driving in racing simulator under strenuous conditions may not be possible for more severe strokes. Therefore, it is important to clarify and acknowledge in the abstract and discussion sections that the results are applicable only in minor stroke and cannot be generally applied to all stroke survivors.

We thank you for this important clarification. We add the qualifier “minor stroke” to lines 27,28, 115, 458, 511, 513, and 566. On lines 552-554, we comment on the possibility for future work exploring more severe stroke.

  1. Figures 1 and 2 look identical and are difficult to interpret. Please revisit and provide adequate explanations for both figures alone with detailed figure captions.

We thank you for this important point. We added additional details to our figure captions (lines 352-357 and lines 399-401). We now include our figures inline to the manuscript, and our submission includes our high resolution figures in the attached zip file.